# Cellulose-Encapsulated Magnetite Nanoparticles for Spiking of Tumor Cells Positive for the Membrane-Bound Hsp70

**DOI:** 10.3390/ijms27010150

**Published:** 2025-12-23

**Authors:** Anastasia Dmitrieva, Vyacheslav Ryzhov, Yaroslav Marchenko, Vladimir Deriglazov, Boris Nikolaev, Lyudmila Yakovleva, Oleg Smirnov, Vasiliy Matveev, Natalia Yudintceva, Anastasiia Spitsyna, Elena Varfolomeeva, Stephanie E. Combs, Andrey L. Konevega, Maxim Shevtsov

**Affiliations:** 1Petersburg Nuclear Physics Institute Named by B.P. Konstantinov of National Research Centre ”Kurchatov Institute”, Orlova Roshcha 1, Gatchina 188300, Russia; dmitrieva_as@pnpi.nrcki.ru (A.D.); marchenko_yy@pnpi.nrcki.ru (Y.M.); deriglazov_vv@pnpi.nrcki.ru (V.D.); smirnov_op@pnpi.nrcki.ru (O.S.); matveev_va@pnpi.nrcki.ru (V.M.); anastasis.8@yandex.ru (A.S.); varfolomeeva_ey@pnpi.nrcki.ru (E.V.);; 2JSC ”Technopark of Saint-Petersburg”, St. Petersburg 197022, Russia; nikolaevhpb@gmail.com; 3Institute of Cytology of the Russian Academy of Sciences (RAS), St. Petersburg 194064, Russia; yakluda5@gmail.com (L.Y.); yudintceva@mail.ru (N.Y.); 4Department of Histology, Saint Petersburg State Pediatric Medical University, St. Petersburg 194100, Russia; 5Department of Radiation Oncology, Klinikum Rechts der Isar, Technical University of Munich, Ismaninger Str. 22, 81675 Munich, Germany; stephanie.combs@tum.de; 6Institute of Biomedical Systems and Biotechnology, Peter the Great St. Petersburg Polytechnic University, Polytechnicheskaya 29, St. Petersburg 195251, Russia; 7NBICS Center, National Research Center “Kurchatov Institute”, Akademika Kurchatova pl. 1, Moscow 123182, Russia

**Keywords:** cellulose microspheres, magnetite nanoparticles, heat shock proteins, membrane-bound Hsp70, TKD-functionalized ferrocellulose, glioma cells

## Abstract

The development of highly sensitive approaches for detecting tumor cells in biological samples remains a critical challenge in laboratory and clinical oncology. In this study, we investigated the structural and magnetic properties of iron oxide nanoparticles incorporated into cellulose microspheres of two size ranges (~100 and ~700 μm) and evaluated their potential for targeted tumor cell isolation. In the smaller microspheres, magnetite-based magnetic nanoparticles (MNPs) were synthesized in situ via co-precipitation, whereas pre-synthesized MNPs were embedded into the larger microspheres. The geometrical characteristics of the resulting magnetic cellulose microspheres (MSCMNs) were assessed by confocal microscopy. Transmission electron microscopy and X-ray diffraction analyses revealed an average magnetic core size of approximately 17 nm. Magnetic properties of the MNPs within MSCMNs were characterized using a highly sensitive nonlinear magnetic response technique, and their dynamic parameters were derived using a formalism based on the stochastic Hilbert–Landau–Lifshitz equation. To evaluate their applicability in cancer diagnostics and treatment monitoring, the MSCMNs were functionalized with a TKD peptide that selectively binds membrane-associated Hsp70 (mHsp70), yielding TKD@MSCMNs. Magnetic separation enabled the isolation of tumor cells from biological fluids. The specificity of TKD-mediated binding was confirmed using Flamma648-labeled Hsp70 and compared with control alloferone-conjugated microspheres (All@MSCMNs). The ability of TKD@MSCMNs to selectively extract mHsp70-positive tumor cells was validated using C6 glioma cells and mHsp70-negative FetMSCs controls. Following co-incubation, the extraction efficiency for C6 cells was 28 ± 14%, significantly higher than that for FetMSC (7 ± 7%, *p* < 0.05). These findings highlight the potential of TKD-functionalized magnetic cellulose microspheres as a sensitive platform for tumor cell detection and isolation.

## 1. Introduction

The isolation and purification of biologically active substances in biotechnology are mainly carried out by column chromatography, which has a number of known limitations on the yield of the target product and the selectivity achieved. Magnetic separation of dispersed carriers is one of the ways to overcome the limitations associated with the need to isolate and diagnose proteins against the background of dominant protein contaminants. The use of carriers in the form of magnetic nano- and microparticles associated with affinity bioligands of the isolated proteins is the way to solve this problem. A commonly accepted method of magnetic affinity separation is the method of covalent binding of the affinant on a magnetic carrier in the technology of direct extraction by incubation in liquid biological media containing the target protein. On the pharmaceutical market one can find patent-protected offers of magnetic separators from Invitrogen Dynal AS, 0379 Oslo, Norway; New England BioLabs (NEB), 240 Country Road, Ipswich, Massachusetts, USA; 2800 Woods Hollow Road, Madison, WI 53711 USA, Promega; 400 Valley Road Warrington PA 18976 USA, Polysciences; Sigris Research, 130 Lilac Ln, Brea, CA 92823, USA, the consumables for which are represented by magnetic microspheres made of cross-linked polymers and other materials [1,2,3]. Magnetic separation has proven itself and found commercial application in the technology of lysozyme extraction from egg white, a number of enzymes such as amylase, galactosidase, pectin, alcohol dehydrogenase, lactate dehydrogenase, phosphatase from cell homogenates of different origins (*Escherichia coli*, *Saccharomyces cerevisiae*, *Bacillus* and placental tissues, pig muscle, etc.) [4,5]. Microspheres with grafted streptavidin from Dynabeads are used as magnetic carriers [6,7,8,9,10]. The most technical solutions in the field of magnetic separation of proteins are distinguished by their original know-how and are usually conducted in laboratory conditions with quality control of the product yield. Furthermore, several nanocarriers-based systems were reported for the efficient separation of tumor from biological samples [11,12,13,14,15,16].

Imparting magnetic properties to microcarriers opens up a number of new possibilities for the directed focusing of target biomolecules in pathological sites, controlled movement of the microcarrier by magnetophoresis and noninvasive monitoring by magnetic resonance imaging (MRI) [17]. Covalent attachment of antigen molecules to a magnetic carrier as part of a single hybrid form allows for the separation of mono- and polyclonal antibodies from cultural biological fluids and the construction of new types of sensitive biosensor elements. The fundamental possibility of obtaining hybrid immunomagnetic forms of the drug allows for combining in a single dosage form a biologically active agent of various natures (antibodies, rDNA, RNA, etc.) and a diagnostic reference product, which is observed by tomographic imaging, fluorescent-optical endogenous detection and directed magnetic focusing in the target organ. Localization of the immunotherapeutic and diagnostic principle in the preparation opens the way for monitoring the fate of the preparation in human using modern instrumental means [18].

Cellulose has been used to fabricate functional cellulose microspheres for biosensing tumor cells using high expression of mHsp70 as a specific indicator. The membrane-associated form of heat shock protein with a molecular weight of about 70 kDa (mHsp70) is a promising marker of malignant cells [19,20,21]. Normal cells are characterized by the expression of Hsp70 at a basic level with localization in the cytoplasm [20,22], while tumor cells are characterized by increased expression and membrane localization of Hsp70 [22,23,24,25,26,27] (reviewed in Shevtsov et al., 2020). An important aspect is the preservation of increased, in relation to normal cells, expression of Hsp70 after epithelial–mesenchymal transition of the cell and the preservation of its high level even in distant metastases [21,28]. With membrane localization of the protein, only a small part of it is exposed to the extracellular space, capable of specific interaction with the peptide TKD developed by Multhoff et al. [29]. The TKD consists of 14 amino acids, which allows of using this peptide as a targeting agent covalently immobilized on a cellulose carrier [30,31].

Preliminary studies have shown the possibility of obtaining microspherical cellulose with internal pores, which has high mechanical resistance during swelling [32,33,34,35,36,37]. The presence of micropores of various sizes provides the possibility of synthesizing MNPs by impregnating cellulose with the corresponding salts of precursors of nucleation of MNPs from iron oxide. The absence of chemical interaction of cellulose with reactants and the absence of swelling makes it possible to synthesize SPM nanoparticles of iron oxide in a microsphere. Magnetic properties of the microspherical cellulose are an important advantage for the functioning of the immunomagnetic carrier, since an external magnetic field allows for its accumulation and fixation in a biological suspension, and the removal of its residues after sorption of the target protein. Subsequently, after removal of the magnetic field, the immunogenic carrier with the target protein can be collected for analysis.

In this study, we investigated the feasibility of producing TKD-functionalized cellulose magnetic microcarriers (TKD@MSCMN) as a specific binding indicator for C6 rat brain tumor cells with high mHsp70 expression and thoroughly characterized them. The MSCMN conjugate with the nonspecific protein alloferron (ALL@MSCMN) served as a control. The TKD activity was confirmed by binding the Hsp70-dye complex to the functionalized ferrocellulose. The ability of both conjugates and non-functionalized MSCMN to bind to mHsp70-expressing C6 tumor cells was assessed in comparison with mesenchymal stem cells.

Ferromagnetism of the microcarriers enables magnetic separation of their complexes with the bound cells from the biological fluid for further studies. The use of functionalized magnetic microcarriers for biosensor diagnostics of tumor cells may prove to be a promising approach in the future. For these purposes, magnetite nanoparticles with a high magnetic moment are preferable. Therefore, an additional objective in this study was to test two variants of Fe_3_O_4_ MNP synthesis: with and without the addition of cesium chloride, which affects the size of the synthesized MNPs. To obtain the cellulose magnetic microcarriers themselves, two synthesis options were also used: (i) the synthesis of MNPs with the addition of cesium chloride was carried out in situ in the pre-prepared cellulose microspheres, using their pores as microreactors; and (ii) with the loading of pre-synthesized CsCl-free MNPs during the synthesis of the ferrocellulose. The composition and structure of the synthesized composite cellulose microspheres with MNPs were studied using XRD, confocal microscopy, and TEM.

To monitor the processes of MNP association directly within the microporous matrix, in addition to microscopy, we used a noninvasive technique of nonlinear longitudinal response to a weak alternating magnetic (*ac*) field parallel to a steady field, adapted for studying MNP ensembles [38]. The dependences of the phase components of the second harmonic of magnetization (NLR-M2) on a steady magnetic field were recorded. The method has high sensitivity and does not require careful preliminary sample preparation [39]. Magnetic and dynamic parameters of the MNPs inside the MSCMNs were obtained from the analysis of the NLR-M2 data using the Fokker–Planck kinetic equation [39].

## 2. Results and Discussion

### 2.1. Characterization of Nanoparticles in Ferrocellulose Microspheres

The structure and composition of MNPs synthesized according to the procedure repeatedly tested earlier [39], before manufacturing the ferrocellulose microspheres (Section 3.1) and after their synthesis inside MSCMN (Section 3.4), were studied by XRD (Section 3.6). The room temperature XRD intensity as a function of the diffraction angle for MNPs synthesized inside the pre-prepared microspheres with the size ~100 μm, as described in Section 3.2 (MSCMN_100), is presented in Figure 1a. In this synthesis a fraction of pre-prepared microspheres with the sizes in the range 80–120 μm was used, as described in Section 3.3. The similar diffraction patterns with different widths of peaks were obtained as well for MNPs themselves (MNP_no-shell), MNPs coated with dextran (MNP_Dx) and ferro-MSCMN with the size ~700 μm (MSCMN_700) synthesized using these ready-made MNPs. To evaluate the size of the MNP crystallinity region, precise treatment of the XRD patterns was performed for all these samples taking into account the instrumental resolution and a doublet structure of the Cu K_α_ line [39,40]. The diffraction peaks broaden, mainly, due to a finite size of the coherent scattering region and internal stress in the sample.

The Williamson–Hall approach differentiates between the size-induced and strain-induced peak broadenings by considering the peak width as a function of the diffraction angle [39,40]:*β_hkl_* cos*θ* = *k*λ/*d* + 4*ε*sin*θ*
where *β_hkl_* is the instrument-corrected breadth (full width at half-maximum) of the *hkl*-reflection located at the angle 2*θ_hkl_*, *d* is the crystallite size, *k* ≈ 0.9, *λ* = 1.54 Ǻ is the wavelength of Cu K_α1_ radiation and *ε* is the strain-induced broadening arising from crystal imperfections and local lattice distortions. The values of *β_hkl_* cos*θ* were plotted as a function of 4sin*θ_hkl_* in Figure 1b for the ferrocelluloses MSCMN_100 (1) and MSCMN_700 (2), as well as for the pre-prepared nanoparticles MNP_no-shell (3) and coated with dextran MNP_Dx (4). From linear fits of the data, the mean sizes *d* and the strain-induced broadening *ε* of the MNPs’ crystallinity region were evaluated and are presented in Table 1. The obtained dimensions of the crystal unit cell were *a* = 0.8381(4) nm and 0.8379(3) nm for MNP_no-shell (3) and MNP_Dx (4), respectively, coinciding within the error limits. Note close values of the crystallinity size obtained for the MNPs synthesized inside cellulose microspheres using CsCl (Table 1) and the MNPs in the colloidal solution synthesized earlier in the same way (9.3(7) nm) [39,41]; their magnetic moments also coincide within the error limits (Section 2.4).

The comparison of the MSCMN_100 and MSCMN_700 parameters from XRD (Figure 1) shows the same magnetite crystalline structure of the MNPs inside them with the smaller width of the diffraction peaks in the latter. This suggests larger crystallinity regions of MSCMN_700, synthesized using the pre-prepared magnetite nanoparticles. These MNPs were synthesized by co-precipitation as in the previous works [39,41] but without adding cesium chloride, resulting, as expected, in the increase in the nanoparticle size.

### 2.2. Confocal Microscope Images

To visualize the spatial shape of the cellulose microspheres and the presence of MNPs inside the composite MSCMN, confocal microscope images were obtained. The images of two different 80–125 μm spherical cellulose microspheres without MNPs and after synthesis of the composite with MNPs, with the diameters 114 and 95 μm, respectively, are shown in Figure 2. The images were obtained in a transmitted light beam. The comparison of the panels (a) and (b) shows higher light absorption in the latter due to MNPs in internal pores of the microsphere.

A confocal image of MSCMN_700 with MNPs and an average size of 700 μm illuminated by quantum dots is shown in Figure 3. As it is seen, high light absorption is observed due to the presence of the MNPs in internal pores. The porous structure of the microspheres is confirmed by uneven distribution of quantum dots observed in the figure. The presence of numerous entrances to the internal pores of the microsphere is clearly visible in Figure 3 as regions without bound quantum dots on its surface and indicates a large number of pores inside.

### 2.3. Transmission Electron Microscopy Results

Figure 4 presents the results of the MSCMN_700 study by TEM. Panel (a) shows a section through a single large ferrocellulose microsphere ~700 μm in size with the pre-synthesized MNPs inside.

On the left edge of the panel, a stripe of accumulated MNPs is observed, probably adsorbed onto the inner surface of the outer shell of the microsphere. Indeed, to the right of this line, individual nanoparticles with sizes of ~17 nm are observed, while to the left of the stripe, they are absent. On the right side of the panel, accumulation areas of smaller lengths are visible. These are probably nanoparticles adsorbed on the cellulose walls of pores inside the microsphere. The panel (b) demonstrates the diameter distribution of MNPs from the panel (a) well described by the lognormal function with the average diameter of 17 nm matching the average crystallinity region found from XRD (17.1 nm) and the width *σ* = 0.29.

### 2.4. NLR-M2 Results

The dependences of the real and imaginary parts of the second harmonic of the magnetization of the nonlinear response *M*_2_ to a weak alternating magnetic field on the scanning field *H* for the suspensions of cellulose microspheres filled with MNPs (MSCMN_100 and MSCMN_700) are shown in Figure 5. The *M*_2_ signals from both samples exhibit a weak *H* hysteresis. The insets present the dependence of the *H*-hysteresis width *H*_c2_ on the scan frequency *F*_sc_. The hysteresis width decreases with the scan frequency due to increasing the time allotted for relaxation of the ensemble of nanoparticles. Such dependence indicates the dynamic nature of the hysteresis inherent to the SPM regime [39,42]. This suggests validity of the mathematical formalism used for approximating the obtained *M*_2_(*H*) dependencies.

The magnetic and dynamic parameters of the ensemble of magnetic nanoparticles in the aqueous suspension MSCMN_100 are presented in Table 2.

From Table 2, the average parameters of magnetic centers change only slightly with the concentration. In colloidal solutions, MNPs may unite in aggregates with the MNP magnetic moments coupled by dipolar forces [39]. Due to this coupling, the magnetic centers from NLR-M2 measurements correspond to the aggregates, if they exist. The question is whether aggregation really occurs or not. The obtained value of *M*_C_ is about 1.8 × 10^4^ μ_B_, close to the average moment 2.5 × 10^4^ μ_B_ of MNPs synthesized earlier in a similar way in the solution without cellulose microspheres [39], suggesting the absence of MNPs’ aggregation and interparticle dipolar correlation in the MSCMNs. This finding can be verified in a different way. From the average MNP size *d* = 10.8 nm measured by XRD, (Table 1), one can obtain the average MNP volume as $V_{c}=\;\pi d_{c}^{3}/6$ = 838 nm^3^ where $d_{c}=dexp(\sigma^{2}/9)$. With the known size of the crystal lattice cell, each containing 24 iron ions, one can determine the average mass of iron in the MNPs as $m_{Fe}=24V_{c}m_{g-at}/a^{3}N_{a}$ = 3.2 × 10^−9^ ng where $m_{g-at}$ = 55.8 is the Fe gram-atom and *N_a_* is the Avogadro number. Using the iron concentration from Table 2, the MNPs’ concentration is determined as *C_Fe_*/*m_Fe_*. The ratio of the latter to the concentrations of magnetic centers obtained from NLR-M2 yields the number of MNPs per aggregate. In Figure 6 the dependence of the amount of MNPs per possible aggregate inside the pores of MSCMN_100 on the concentration of Fe in the suspension is presented. From the figure, the average amount of MNPs per possible aggregate is, within the errors, close to one, which confirms the absence of aggregation.

The samples containing MSCMN_700 with the number of microspheres from one to one hundred synthesized using pre-prepared MNPs were examined at the NLR-M2 device as well. The moduli of the integral recorded signals normalized by *h*^2^ for the scan frequencies 0.25 and 10 Hz are shown in Figure 7. The linear dependence demonstrates an additive character of the signal. The inset shows the ratios of the square of the hysteresis loop to the modulus of the *M*_2_ signal. This quantity is almost independent of the number of microspheres. Together with the linear increase in the integral signal this indicates the absence of any magnetic coupling of the microspheres. The increase in the *H-*scan period from 100 ms to 4 s is accompanied by the decrease in the hysteresis by two times, suggesting dynamic nature of the hysteresis and, hence, the single-domain state of the MNPs inside the microspheres.

As can be seen from the TEM image in Figure 4a, some of the nanoparticles in large Cel microspheres are in a colloidal solution in the pores, while the other part forms striped structures on the porous walls. This may affect the magnetic measurements via dipole–dipole coupling of the wall MNPs. The parameters obtained by fitting the signals from 1, 10 and 20 MSCMN_700 are presented in Table 3. The magnetic moment (and volume) distribution width *σ_M_* ≈ 0.86 is found to be, expectedly, three times the diameter distribution width *σ_d_* obtained from TEM (Figure 4b). This finding evidences no noticeable effect of the dipolar coupling between the MNPs adsorbed on the walls and, thus, the identity of all MNPs in the microspheres.

Unexpectedly, the average magnetic moment of MNPs in MSCMN_700 obtained from the magnetic measurements (Table 3), turned out to be smaller (~1.4 × 10^4^ μ_B_) than the magnetic moment of MNPs in MSCMN_100 (~2 × 10^4^ μ_B_, Table 2), despite the larger size of the former (Table 1). This results from considerable microcrystal strains suppressing the MNP magnetism. The decrease in the nanoparticle magnetization down to 25% and lower compared to the bulk value was also observed in a number of MNPs, including magnetite, with microcrystal strains of the same order *ε*~10^−3^ [43]. Unstrained bulk magnetite has cubic crystal symmetry with the magnetic moment directed along the main diagonal of the unit cell. MNP microcrystal strains distort the local symmetry, resulting in redirecting the easy axes of magnetization. These uncorrelated lattice distortions partially disorder local magnetic moments, decreasing the total moment of a nanoparticle. For the four samples studied, with occasionally different microcrystal strains, this effect is presented in Figure 8 via the dependence of the normalized mean MNP magnetic moment $M_{c}/M_{c}^{max}$ on the factor $R=E_{a}^{max}/U.$ Here, the magnetic moment $M_{c}$ was obtained from the NLR-M2 measurements and $M_{c}^{max}$ is the estimated magnetic moment corresponding to the mean volume $V_{c}=\;\pi d_{c}^{3}/6\;$, but with the magnetization of single-crystal magnetite at room temperature 0.08 Am^2^/g [43]. The corresponding diameter is $d_{c}=dexp(\sigma_{d}^{2})$, with *d* estimated from the XRD measurements and the size distribution width $\sigma_{d}=\sigma_{V}/3$, with $\sigma_{V}$ obtained also from the NLR-M2 measurements. Note that the so obtained normalized magnetic moment is somewhat underestimated, as $M_{c}^{max}$ was assessed without account of the demagnetization effect.

The disordering of local magnetic moments of a nanoparticle is determined by the relationship between the energies of microcrystal strains and anisotropy. The energy of elastic deformation for a nanoparticle reads $U=EV_{c}\varepsilon^{2}/2$, where *E* is the Young modulus, which for magnetite is approximately 200 GPa. The anisotropy energy to be compared to U is ${E_{a}^{max}=M}_{c}^{max}H_{a}/2$, where $H_{a}$*=* 680 Oe is the anisotropy field of single-crystal magnetite [43]. The ratio $R=E_{a}^{max}/U$ is used here as a measure of competition between the two energies. From the figure, the normalized magnetic moment is well fitted by the function $M_{c}/M_{c}^{max}=1-e^{-x}$ where $x=\xi R={\left(\varepsilon_{0}/\varepsilon \right)}^{2}$ with ξ = 0.736 and $\varepsilon_{0}=0.322\;\times \;10^{-3}$ in the range *R* > 0.1. The tendency breaks when $U\gg E_{a}^{max}$. In this phenomenological function, the strain energy *U* disordering local magnetic moments is somewhat similar to temperature in the Boltzmann factor playing the same role. This finding ensures that microcrystal strains are the main factor diminishing the magnetic moment of nanoparticles. Thus, XRD can be used for preliminary testing of the magnetic quality of MNPs to be applied in biomedicine, $\varepsilon_{0}$ being the measure of the quality.

### 2.5. Functionalization of Ferrocellulose Microspheres

TKD-peptide providing selective binding to the HSP70 expressed in the cytoplasmic membrane was chosen for functionalization of the ferrocellulose microspheres (TKD@MSCMN), described in Section 3.11. It is important to note that the part of Hsp70 responsible for the specific interaction with TKD is exposed to the extracellular space of the C6 cells [44]. The functionalized MSCMN_700 were then used, as they have a larger average size and, accordingly, a larger surface area for coupling with the cells compared to MSCMN_100 later on, designated as MSCMN for simplicity. As a negative control for in vitro experiments, a conjugate of the alloferon peptide (no selective binding with the C6 membrane) with the magnetic microsphere cellulose (All@MSCMN) was used.

The conjugate formation and the localization of the TKD peptide were determined using the fluorescently labeled TKD-FITC peptide for conjugation. Figure 9 shows the fluorescence microscopy of TKD-FITC@MSCMN before (panel (a)) and after (panel (b)) mechanical disruption. The images obtained demonstrate immobilization of the peptide on the carrier. Its surface and internal localization are evident from the green fluorescence from the entire surface of the MSCMN both before and after disruption. Mechanical destruction leads to an increase in the fluorescence intensity, which indicates the accumulation of the peptide not only on the outside, but also in the depth of the porous carrier.

Information on the localization of the TKD peptides and the relative surface area of MSCMN occupied by them after the functionalization and washing off the free peptides can be obtained from the experimentally determined mass 358 μg of the peptides bound to 1 g of MSCMNs. The mass of a single MSCMN with a diameter of 700 μm filled with water is 180 μg. Therefore, 1 g of MSCMNs contains 5555 particles. The molecular mass of the TKD peptide is 1500, from which we find that 2.39 × 10^−7^ mol of TKD are bound to 1 g of MSCMNs, and, accordingly, 2.6 × 10^13^ peptides are bound to one MSCMN. The TKD peptide has a beta-sheet configuration [31,45]. This suggests that the contact area of the peptide is approximately 20 nm^2^, since the length of the peptide bond in the beta-sheet is about 4.5 nm [31,45]. The total area of all peptides bound to one MSCMN is 5.2 × 10^6^ μm^2^, which significantly exceeds the surface area of the MSCMN of 1.5 × 10^6^ μm^2^. This indicates the localization of the peptides both on the surface and inside the MSCMN in accordance with Figure 9b, which shows a significant increase in TKD-FITC fluorescence upon destruction of the MSCMN. Comparison of the volumes of the MSCMN (1.8 × 10^8^ μm^3^) and all the peptides bound to it (~2.1 × 10^8^ μm^3^) confirms this result.

### 2.6. Fluorescence Microscope Study of Ferrocellulose Complexes with Hsp70

The biological activity of the TKD peptide in the TKD@MSCMN conjugate after immobilization was verified using co-incubation of the conjugates with Hsp70 stained with Flamma648, (see Section 3.12). Visualization was performed in the red channel of a fluorescence microscope and is shown in Figure 10. The signal from the target conjugate presented in panel (b) demonstrates the presence of Hsp70 in the complex. The sorption of Hsp70 by the control conjugate with the ALL peptide was absent, as is seen in Figure 10a.

The obtained microscopic images were processed in ImageJ 1.54g code. During the analysis, the area of the sections whose fluorescent signal exceeded the preset threshold was determined. To obtain the normalized values *S_norm_* of the stained area, the ratio of the total area of the stained fluorescent sections to the area of the MSCMN_700 was calculated. The results are presented in Figure 11.

The statistical significance of the obtained differences in the binding of the Hsp70-Flamma648 complex to the control and target conjugate was tested using Student’s *t*-test with an accepted significance level of 0.05 and showed statistically significant differences.

### 2.7. Fluorescence Microscope Study of Ferrocellulose Complexes with C6 and FetMSC Cells

Figure 12 shows fluorescent images of the ferrocellulose complexes with Hsp70-overexpressing C6 cells stained with the vital nuclear dye Hoechst 33342 as described in Section 3.13. The visualization results demonstrate the sorption of the cell culture on the surface of the carrier, as evidenced by the characteristic blue fluorescent signal of the Hoechst 33342 from the cell nuclei. The comparison of the image from the TKD@MSCMN complex with C6 cells (Figure 12b) with the control images from the complexes: (i) of All@MSCMN with C6 cells (panel (a)); (c) of TKD@MSCMN with FetMSC cells; and (d) of the non-functionalized ferrocellulose with C6 cells shows statistically significant excess in the number of sorbed cells in the first. This result confirms the binding specificity of the TKD-peptide-functionalized MSCMN to mHSP70 on the C6 cell membrane. A graphical representation of the calculated values of the fluorescently stained areas normalized to the area of the ferrocellulose is shown in Figure 13.

The groups were compared using Student’s *t*-test with a significance level of 0.05, which revealed statistically significant differences between the cell sorption on TKD@MSCMN and ALL@MSCMN as well as on the non-functionalized MSCMN, indicating selectivity of the cell sorption on the target conjugate.

The larger average size of MSCMN suggests better characteristics of this type of ferrocellulose for performing the biomedical diagnostic tasks formulated in the work. Indeed, the larger size of the MSCMN increases the area of interaction with C6 cells and, accordingly, the probability of their binding. The comparisons of the binding results for the Hsp70-dye complex and C6 glioblastoma cells, as well as for normal FetMSCs showing lower mHSP70 expression, with MSCMN_700 and MSCMN functionalized with alloferron and TKD are shown in Figure 10, Figure 11, Figure 12 and Figure 13. The data evidence more efficient binding of Hsp70-dye compared to the C6 cells, showing hyperexpression of mHSP70. As shown in Figure 9, after incubation the TKD peptide accumulates both on the surface of MSCMN_700 (Figure 9a) and in its pores (Figure 9b). The spatial structure of MSC is represented by a network of pores. Transport pores have a size of 0.1–3.0 μm; they provide instant swelling of cellulose. Narrow pores (chromatographic), which determine the capillary properties of the carrier and implement constant exchange of the “internal” and “external” solvent, have a size of 5–30 nm [36]. Therefore, both the TKD-peptide at conjugation and Hsp70-dye complex at incubation with conjugate can penetrate through the pores inside the MSCMN_700. The C6 cells have dimensions of ~5–10 μm [46], larger than the pore size in Cel-m_spheres, and they bind to the TKD peptide localized on the surface of the ferrocellulose. The cells can be further extracted for analysis by removing the complexes from multi-component solutions with a constant magnetic field.

Important information for assessing the tumor cell isolation is the number of cells per ferrocellulose globule. In these experiments, the cell nucleus is stained with DAPI. The ratio of the cells to MSCMN_700 (as well as TKD@MSCMN_700 and ALL@MSCMN_700) during coincubation was 10^6^/100, as described in Section 3.13. Using the fluorescence microscope images of the C6 complexes with the TKD@MSCMN and ALL@MSCMN conjugates (panels (a), (b) in Figure 12), we determined the square and size of a nucleus of one C6 cell, *D* ≈ 7.2 μm, matching the known size of the C6 nucleus, 6–8 μm. By determining the area occupied by DAPI in the image of the TKD@MSCMN complex with C6 cells (Figure 12a) and dividing it by the area of a single nucleus, we found the number of the adsorbed cells on the visible half of the complex. Doubling this number allowed us to estimate the number of the adsorbed cells on the entire surface of TKD@MSCMN_700. After 10 repetitions, the estimate was 5400 ± 300 C6 cells on MSCMN_700. Based on the data obtained for the FetMSC cells, the sorption of these cells on MSCMN_700 appeared to be approximately four times lower.

Potential development of the presented magnetic affinity approach on base of mHsp-targeted species involves expanding the laboratory application of magnetic separation to real biological fluids, such as plasma, blood, and urine. The magnetic affinity separation of mHsp-labeled analytes from blood obtained from breast cancer patients may be the next step toward medical application. The advantages of the discussed approach, such as sensitivity to various cancer types expressing mHsp in the cell membrane, rapid flow-through extraction, and operational safety, can be exploited in efferent therapy. Cancer is known to cause local tissue inflammation with the release of endotoxins and proinflammatory cytokines. Potential hydrophobic modification of cellulose microspheres can be used to remove endotoxins during hemodialysis. When combined with a magnetic biosensor for binding mHsp-positive cells, this will enable the simultaneous study of the mHsp-positive cell and endotoxin levels, an important and promising area in early cancer diagnosis. Construction of portable magnetic assay systems combined with the standard dialysis apparatus for detoxication and blood correction is the perspective way of magnetic microbead material. The last perspective area is to develop the facility for isolation of bioactive substance associated with the Hsp protein from the media in recombinant biotechnology for personalized medicine.

## 3. Materials and Methods

The synthesis of hybrid composites of iron oxide magnetic nanoparticle with microspherical cellulose was carried out using two approaches [47]: (i) the synthesis of NSCMN by direct administration of pre-prepared MNPs (Section 3.1) to the solution of completely dissolved microcrystalline cellulose in heated calcium thiocyanate after pouring it into heated industrial oil I-8 with stirring (Section 3.2) and (ii) alternatively, by precipitation from the solutions of iron salts (II) and (III) directly in the pores of the preformed microspherical particles of cellulose.

### 3.1. Synthesis of Iron Oxide Nanoparticles

Iron oxide nanoparticles were obtained by coprecipitation from a solution of di- and trivalent iron salts (1:2) in an alkaline medium [39,41]. The precipitation reaction was carried out at 80 °C under a nitrogen atmosphere with vigorous stirring. Cesium chloride was used to reduce the size of the resulting nanoparticles, and ammonium hydroxide served as a precipitant. The black precipitate formed during the reaction was separated with a magnet and washed several times with distilled water until a neutral pH was achieved. The resulting suspension was stored in a tightly sealed container at 4 °C.

The synthesis of MNPs was carried out without adding cesium chloride, assuming an increase in the magnetic core size. Some of the MNPs after synthesis were coated with the dextran shell to compare their parameters with the parameters of MNPs “coated” by dextran synthesized using CsCl, which we had widely used previously in biomedical applications.

### 3.2. Synthesis of Magnetic Microspherical Cellulose Using Pre-Prepared Iron Oxide Nanoparticles

The first method is designed to obtain hybrid composites from magnetic nanoparticles and micronized cellulose capable of conjugating bioligands after activation of the chemical groups of the polysaccharide [47].

The synthesis setup consists of a chemical reactor with a housing, a stirrer with adjustable rotation speed, and a thermostat. The process was carried out in a cylindrical reactor with a volume of 0.5 L with heating at a temperature of up to 110 °C. Microcrystalline cellulose was added to an aqueous solution of calcium thiocyanate with a density of 1.40(1) g/mL (20 °C) at a rate of 6 g per 100 mL and left for two hours for complete swelling. Then, the sorbed air was pumped out of the mixture using a water-jet pump. The resulting suspension was heated to 105 °C until the cellulose was completely dissolved and poured into heated I-8 industrial oil in the volume ratio 3:1 of oil:cellulose. Then, iron oxide nanoparticles were added to the solution, and spherical microspheres with the embedded nanoparticles were formed over two hours with constant stirring at 105 °C. After completion of the process, the reaction mixture was cooled with running water for 2 h; the resulting particles were precipitated and the oil was drained. Excess oil was washed out with ethanol in the Soxhlet apparatus. The required particle fractions were isolated on sieves. The schematic representation of the synthesis of this type is shown in Figure 14a.

The concentration of MNPs in MSCMN was estimated by the thiocyanate method as described in Section 3.5.

### 3.3. Preparation of Microspherical Cellulose

To obtain microspherical cellulose in the technological process described in Section 3.2, after the stage of pouring the microcrystalline cellulose completely dissolved in calcium thiocyanate into hot industrial oil I-8 for two hours at 105 °C and continuous stirring, the spherical microspheres formed.

After completion of the process, the reaction mixture was cooled with running water for 2 h. Then, the obtained particles were precipitated and the oil was decanted. Excess oil was washed with ethanol in the Soxhlet apparatus. Three fractions of particles with sizes of 10–30, 40–80 and 80–125 μm were separated on sieves. Particles in the suspension had the form of porous microspheres with the external diameter in the range 10–200 μm. The total volume of capillary pores was >90%, which was determined in scanning electron microscopy and NMR studies of microcellulose [47].

### 3.4. Magnetic Microspherical Cellulose with the Synthesis of Iron Oxide Nanoparticles Inside Pre-Prepared Cellulose Microspheres

The second, alternative, method for obtaining the hybrid magnetic microsphere cellulose, which reduces possible aggregation of iron oxide nanoparticles during thermal precipitation of the cellulose particles from the solution, is the reaction of the nanoparticle formation in the pre-formed microspherical porous cellulose [47].

The microspherical porous cellulose with a particle size of 80–125 μm and an internal pore volume of more than 90% was treated with a mixture of 0.1 M iron sulfate and 0.2 M iron (III) chloride. The concentration ratio of the solutions was 1:2. Iron oxide was precipitated by treating the cellulose matrix impregnated with the iron salts with the 2 N aqueous ammonia solution. The process was carried out at 80(1)°C for 4 h. As a result, a hybrid composite of nanoparticles with dark-brown cellulose was obtained. The schematic representation of the synthesis of this type is shown in Figure 14b.

### 3.5. Determination of the MNPs’ Concentration by the Thiocyanate Method

The iron concentration in an aqueous suspension of magnetic microspherical cellulose was determined colorimetrically using the thiocyanate method based on the optical density of the colored iron-thiocyanate complex formed by the interaction of Fe^3+^ ions with potassium thiocyanate [48]. The procedure included the following steps: the microspheres were dissolved in concentrated nitric acid by heating in a water bath (80 °C, 10 min) to convert the iron to the ionic form Fe^3+^, then the solution was cooled to room temperature and diluted with distilled water, after which the sample was mixed with a 0.8 M solution of KSCN. The optical density of the sample was measured 1 min after mixing using a Genesys 50 spectrophotometer (Thermo Fisher Scientific Inc., Waltham, MA, USA) at a wavelength of 575 nm relative to a control aqueous solution of non-magnetic microspherical cellulose. The quantitative iron content was calculated using a calibration curve constructed using standard solutions of ferroammonium alum.

### 3.6. Structure Examination of Iron Oxide Nanoparticles by X-Ray Diffraction Inside MSCMN

The diffractometer Rigaku Holdings Corporation (Tokyo, Japan) providing Cu K_α_ line radiation with the wavelength λ = 1.5405 Å used.

### 3.7. Confocal Microscope Measurements of Ferrocellulose

The images were obtained with an inverted confocal laser-scanning microscope LSM 510 META (Carl Zeiss, Jena, Germany) and processed with the software that came with the microscope.

### 3.8. Transmission Electron Microscopy Study

The sample with ferrocellulose microspheres of the larger size ~700 μm was studied using the electron microscope JEM-100C (Jeol, Tokyo, Japan). The microspherical cellulose particles were embedded in Epon resin. Ultrathin sections of 100 nm were prepared for TEM.

### 3.9. NLR-M_2_ Study of Ferrocellulose

The magnetic measurements were performed with the homemade setup [42,49] consisting of two *dc*-field Helmholtz coils, a radiofrequency generator of the *ac* magnetic field *h* with a low-frequency filter at the output, a two-mode (*f*, 2*f*) resonant sensor with a sample holder, and a receiver registering the *2f* response signal with a high-frequency input filter. Particular technical solutions ensured the high sensitivity 10^−10^ emu of the device, such as the presence of a two-mode sensor; effective recording the output nonlinear response signal with a 2*f*-mode selective system; deep suppression of the spurious 2*f* voltage from the generator at the input of the two-mode sensor and the *f* voltage at the input of the receiver by high- and low-frequency filters, respectively; a large range of the *ac-*field amplitude up to 15 Oe with keeping low noise at the receiver input in the whole range; and the use of the elements and materials in the two-mode sensor not generating a spurious 2*f* signal. The *Q*-factor of the second harmonic *Q*_2_ ≈ 200 enhances the setup sensitivity by the factor √*Q*_2_ compared to the non-resonance *ac* susceptibility measurements.

Both real and imaginary components of the *M_2_* response were recorded simultaneously as functions of the magnetic field *H* slowly scanning in the range from −300 to 300 Oe and backwards symmetrically to zero *H*. The sample temperature was stabilized close to room temperature by a flow thermostat using evaporated nitrogen. Two *H-*scan frequencies, 0.25 and 10 Hz, were used to control the field hysteresis of the signal, if it exists. The scan frequency effect on the hysteresis would mean its dynamical character pointing on the SPM nature of the magnetic centers [42,49]. The condition *h* << *H* was accomplished in the major part of the *H*-field region, ensuring validity of the second-order susceptibility obtained in the perturbation theory [39,41,47] for qualitative analysis of the experimental data. This technique showed itself to good advantage in studying biodistribution of the dextran coated MNPs functionalized with monoclonal antibodies to the heat shock protein HSP-70 in C6 glioma rats [50].

### 3.10. Processing NLR-M2 Experimental Data

Quantification of the NLR-M2 data obtained was executed by processing the dependences Re*M*_2_(*H,T*) and Im*M*_2_(*H,T*) with the formalism containing a numerical solution of the Fokker–Planck equation describing kinetics of SPM particles. This approach had been effectively applied in the study of colloidal aqueous solutions of dextran-coated MNPs [33] and biodistribution of mesenchymal stem cells in animal models [41].

The computational resources of PIK Data Processing Centre of NRC “Kurchatov Institute”, PNPI (Gatchina, Russia), were involved, with self-made software. Prior to the fitting, the raw data were averaged between the direct and reverse scans and the antisymmetric part relative to zero *H* was extracted as required by the theory. The following parameters widely characterizing magnetic properties of the MNP ensembles were obtained: the lognormal distribution width and the average value of magnetic moments, the saturation magnetization, the concentration of magnetic centers proportional to the saturation magnetization [39,41], and the magnetic anisotropy, as well as the parameters of magnetization dynamics, viz., the damping factor and the longitudinal relaxation time.

If the signal-to-noise ratio of the *M*_2_ response is poor, significant errors may emerge in the parameters obtained in the course of processing the signal with the formalism. As *M*_2_~*h*^2^, the moduli of the integral *M*_2_ signals normalized to *h*^2^ can be used for relative assessment of the content of magnetic centers in different samples with the similar magnetic moment distributions. In particular, one can estimate the content of MNPs uploaded to cellulose microsphere during the synthesis comparing their *M*_2_ signals with that of the suspension of pre-synthesized MNPs with the known concentration of Fe [42]. This approach allows for obtaining a more reliable estimate of the concentration of magnetic centers in the samples with small iron contents.

### 3.11. Functionalization of Magnetic Microspherical Cellulose

TKD-peptide is represented by a sequence of 14 amino acids—TKDNNLLGRFELSG —and has a molecular weight of 1.5 kDa. It provides selective binding to the HSP70 protein, increased expression of which in the cytoplasmic membrane is characteristic of particularly malignant solid tumors prone to invasion and metastasis. The structural morphology of the protein does not change with its membrane localization. Therefore, the latter was chosen for functionalization of the ferrocellulose microspheres.

Before conjugation with the peptide, the surface of the MSCMN was modified with amino groups [51]. The amination process includes three main stages: (i) activation of the hydroxyl groups of cellulose with a sodium hydroxide solution, (ii) introduction of reactive epoxy groups during the interaction of alkoxide anions (–O^−^) with epichlorohydrin and (iii) addition of amino groups to the formed epoxy groups by introducing ammonium hydroxide into the reaction mixture. Dioxane, an aprotic polar solvent, was used to prevent particle aggregation during amination. Aminated MSCMN-NH_2_ was stored under a layer of distilled water.

Conjugation of the TKD peptide with MSCMN-NH_2_ was carried out using the carbodiimide method [52]. Compounds containing a carbodiimide group act as activators of carboxyl groups in amino acids, which allow for the formation of peptide bonds, cross-linking of amino acid fragments, or conjugation. The amide bond results from the condensation reaction between COOH and NH_2_ groups. The use of carbodiimide allows for the reaction to be carried out under mild conditions (room temperature, neutral pH of the medium, ambient pressure) important for preventing peptide denaturation. Quantitative determination of the TKD peptide in the TKD@MSCMN conjugates was performed using the Bradford method [53]. To verify the formation of the conjugate and to establish the localization of the TKD peptide during its covalent immobilization using the carbodiimide method, conjugation of MSCMN-NH_2_ with the fluorescently labeled peptide TKD-FITC (CF) (Carboxyfluorescein-TKDNNLLGRFELSG, EMC, Leverkusen, Germany) was carried out. Microscopy was performed on a ZOE fluorescence microscope. The signal from the FITC dye was recorded in the green channel.

The peptide alloferon (ALL), consisting of 13 aminoacids—HGVSGHGQHGVHG—(State Research Institute of Highly Pure Biopreparations, Saint Petersburg, Russia) was used as a negative control for in vitro experiments. The conjugate of alloferon with magnetic microsphere cellulose (All@MSCMN) was synthesized using the method described above in this section.

The conjugates were stored at 4 °C; to avoid contamination during storage, the NaN_3_ solution (0.04%) was added to the conjugates.

### 3.12. Control of TKD Peptide Activity in the Conjugate

The activity of the TKD peptide in the conjugate was monitored by studying the specific sorption of the Hsp70 protein on TKD@MSCMN-NH2 after co-incubation of the stained Hsp70 with the conjugate by measuring the dye fluorescence. The All@MSCMN-NH2 conjugate was used as a negative control. For covalent staining of Hsp70, the Flamma648 protein labeling kit (BioActs, Strasbourg, France) was selected. Co-incubation was carried out on an orbital shaker for 60 min (600 rpm), adding 10 μL of a solution containing Hsp70-Flamma648 to the MSCMN samples, the concentration of which during incubation was 1.04 mg/mL for protein. Then, the supernatant was removed, and two consecutive washes were performed with a phosphate-buffered saline solution. Microscopy was performed on a ZOE fluorescence microscope. The signal from the Flamma648 dye was recorded in the red channel.

### 3.13. Cultivation of Glioma Cells of C6 Line and Mesenchymal Stem Cells with MSCMN and Conjugates

Human fetal mesenchymal stem cells (FetMSCs) were cultured in DMEM medium (Sigma-Aldrich, St. Louise, MO, USA). The medium was supplemented with 10% fetal bovine serum, 100 IU/mL penicillin, and 100 μg/mL streptomycin (Gibco, St. Louis, MO, USA). The cultivation was performed under standard conditions (37 °C, 5% CO_2_) with regular changing of the culture medium. The fetal mesenchymal stem cells isolated from human bone marrow were obtained from the Collective Use Center “Collection of Vertebrate Cell Cultures,” funded by the Ministry of Science and Higher Education of the Russian Federation, at the Institute of Cytology of the Russian Academy of Sciences (St. Petersburg, Russia).

The study used the C6 cell line (rat glioma) obtained from the Collective Use Center “Collection of Vertebrate Cell Cultures” of the Irkutsk Scientific Center of the Russian Academy of Sciences. The choice of the C6 cell line was due to the increased level of mHsp70 expression. In this regard, selective binding of the TKD@MSCMN conjugate to the mHsp70 expressed on the cell surface is possible.

Preparation of cell cultures

Both types of cells were cultured under standard conditions using the DMEM/F12 nutrient medium (BioInLab, Rostov, Russia) containing 10% fetal bovine serum (FBS), (Gibco, Logan, UT, USA) and gentamicin (10 mg/mL) (BioInLab, Rostov, Russia) at 37 °C in an atmosphere of 5% CO_2_. To obtain the cell suspension, they were treated with the 0.25% trypsin/EDTA solution (Gibco, Logan, UT, USA). The cells were counted and stained with the vital nuclear dye Hoechst 33342 (Lumiprobe, Moscow, Russia). The dye was added to the cell suspension at a ratio of 1:1000.

2.Incubation of cells with MSCMN samples:

The suspension of the cells with a concentration of 10^6^ mL^−1^ and 100 MSCMNs or 100 functionalized MSCMNs was incubated for 2 h at 37 °C on a thermoshaker (150 rpm).

Incubation was performed with the following samples:(1)TKD@MSCMN—target conjugate (TKD peptide covalently immobilized on ICMN);(2)All@MSCMN—negative control (ICMN conjugated with a non-specific peptide);(3)MSCMN without any peptide—the control sample.

After coincubation of the cells and microcarriers, the supernatant was removed, and the MSCMN samples with the adsorbed cells were resuspended in 1 mL of PBS and transferred to glass slides. Fixation was performed with 100 μL of 10% formalin for 10 min at room temperature. The formalin was then removed, washed with PBS, and the slides were mounted using Mounting Medium (VECTASHIELD, Vector Laboratories, San Ramon, CA, USA), which prevents fluorescence fading and ensures long-term storage of the slide. A coverslip was then applied and the slides were sealed.

### 3.14. Fluorescence Microscope Study of Ferrocellulose Complexes with C6 and MSC Cells

The manufactured fixed preparations containing the ferrocellulose complexes with C6, FetMSCs cells and with non-functionalized MSCMN were analyzed using a Bio-Rad ZOE™ fluorescence microscope. The nuclear dye Hoechst 33342 was excited by a UV LED at a wavelength of 355/40 nm. The cells adsorbed on the surface of the microcarriers were evaluated.

### 3.15. Statistical Analysis

Statistical processing of results for *n* = 10 was performed with GraphPad Prism 8.4.3. After testing for normal distribution (Shapiro–Wilk test) and the absence of outliers, the data were analyzed using Student’s *t*-test. Results are expressed as mean ± 95% CI; differences were considered significant at *p* < 0.05.

## 4. Conclusions

This study demonstrated the feasibility of producing cellulose-based magnetic microspheres functionalized with the TKD peptide for selective binding to C6 tumor cells. The bioassay relies on the recognition of membrane-bound mHsp70, overexpressed on tumor cells, by the TKD peptide. The resulting TKD-functionalized ferrocellulose microspheres efficiently recognized and bound to 5400 C6 tumor cells per one TKD@MSCMN_700 globule due to its large surface area. The magnetic moment of the microspheres ensures reliable magnetic isolation of mHsp70-expressing cells. This indicates that TKD-functionalized magnetic cellulose is a promising affinity ligand for modern commercial and laboratory systems for the detection and isolation of tumor cells from biological fluids. Magnetic functionality was achieved both by embedding pre-synthesized superparamagnetic magnetite nanoparticles into the cellulose matrix and by synthesizing them in situ. The nanoparticles were synthesized with and without cesium chloride, which influenced the particle size. X-ray diffraction with the Williamson–Hall analysis and the second-harmonic magnetization measurements with the treatment based on the Fokker–Planck formalism allowed us to characterize the MNPs’ structural and magnetic properties. Although the exclusion of CsCl resulted in larger nanoparticles, they exhibited increased microcrystalline strain and reduced magnetic moments. In contrast, the use of CsCl resulted in smaller MNPs of structurally higher quality and with stronger magnetic moments. The information obtained is important for the synthesis of MNPs for use in MRI diagnostics and tumor therapy using hyperthermia and radiosensitization.

## Figures and Tables

**Figure 1 ijms-27-00150-f001:**
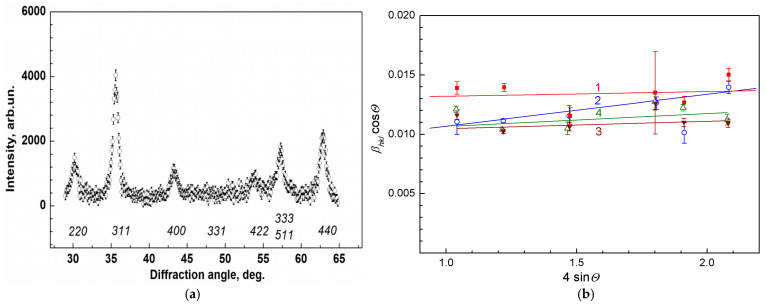
Panel (**a**) presents X-ray diffraction intensity vs. diffraction angle for ferrocellulose microspheres with the sizes ~100 μm. The nominal reflections for magnetite are presented below the peaks. Panel (**b**) displays the treatment of the data with Williamson–Hall formula for: (1) MSCMN_100 (red filled squares and line); (2) MSCMN_700 (blue open circles and line); (3) MNP_no-shell (open green triangles) and (4) MNP_Dx (filled brown reverse triangles), respectively.

**Figure 2 ijms-27-00150-f002:**
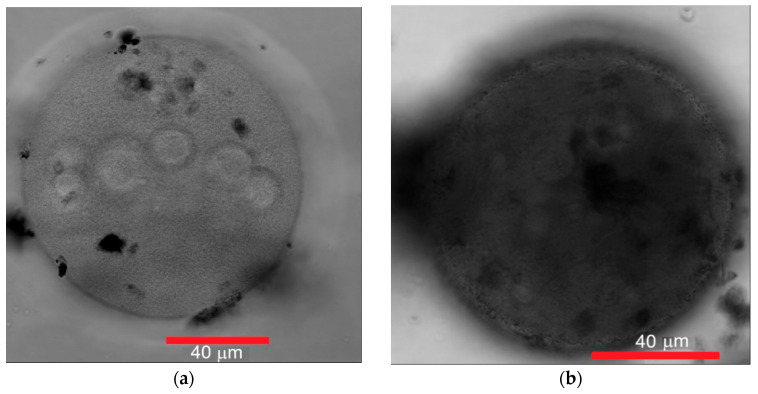
Confocal images of two different spherical cellulose microspheres with sizes in the range 80–125 μm without MNPs (**a**) and with MNPs synthesized inside the microsphere (**b**).

**Figure 3 ijms-27-00150-f003:**
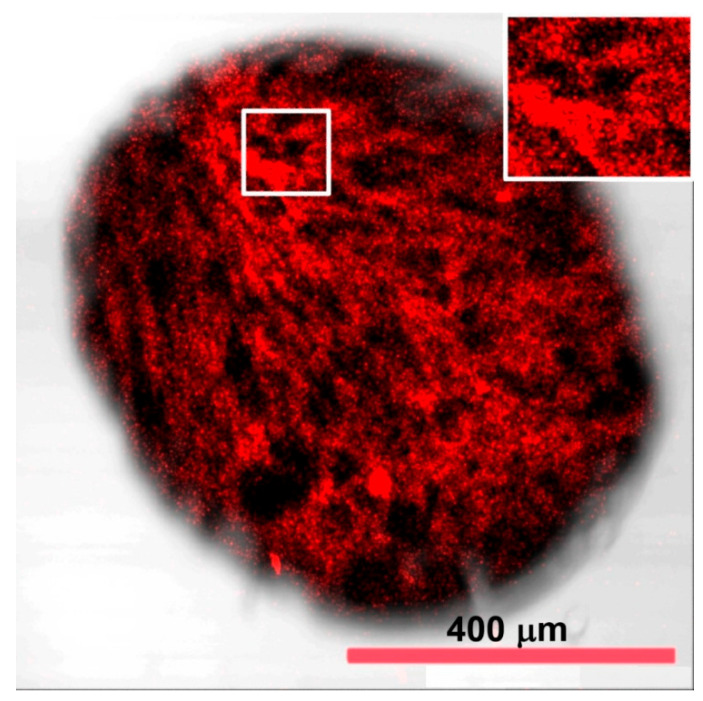
A confocal image of MSCMN_700 with pre-synthesized MNPs. Quantum dots with emission in red and infrared wavelength ranges indicate the presence of pores inside.

**Figure 4 ijms-27-00150-f004:**
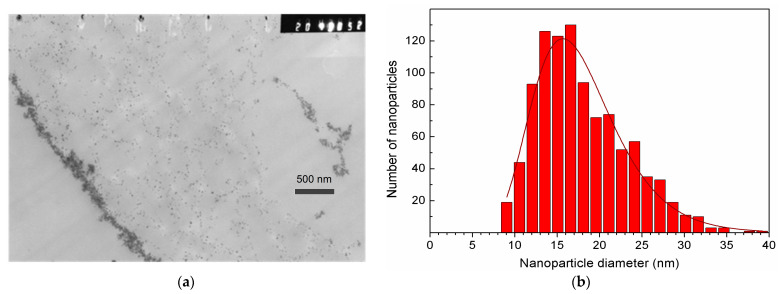
TEM image of a section of the MSCMN_700, (**a**). The measured distribution of MNP diameters inside the microsphere with a statistical sample of *n* = 1000 (histogram) and its best fit to the lognormal distribution (solid line) (**b**).

**Figure 5 ijms-27-00150-f005:**
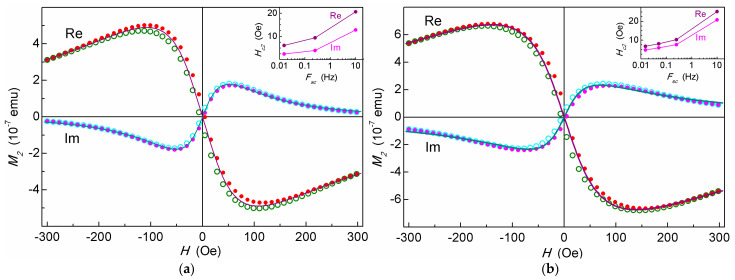
Real and imaginary parts of *M*_2_ response as functions of *dc* magnetic field from suspensions of MSCMN_100, *C_Fe_* = 503 μg/mL (**a**) and from the sample with 15 MSCMN_700, *C_Fe_* = 16.6(4.4) μg/mL (**b**) in water at room temperature registered at *F*_sc_ = 0.25 Hz and *ac-*field amplitudes of 2.8 Oe and 14 Oe, respectively. The solid curves are best fits. Filled and open symbols present direct and reverse *H* scans, accordingly. Every 40th point is shown. Insets show the dependences of the field hysteresis width (“coercive force” *H*_c2_) for Re*M*_2_(*H*) on the *H*-scan frequency.

**Figure 6 ijms-27-00150-f006:**
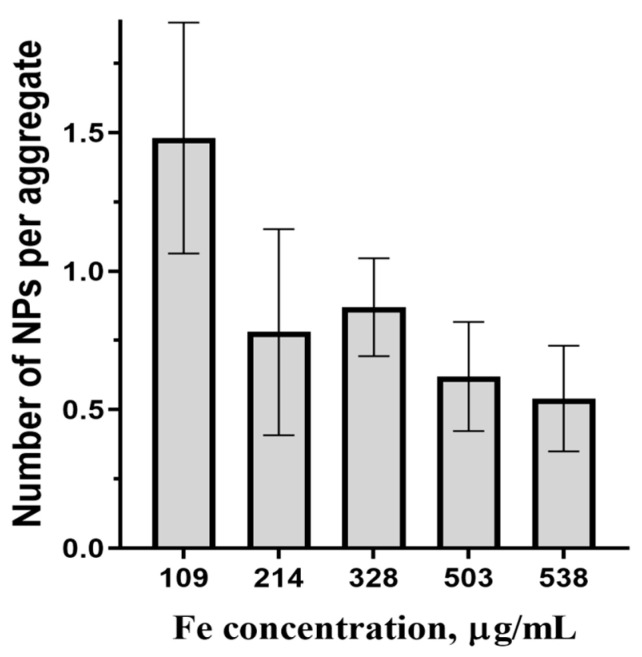
Average number of MNPs per aggregate in MSCMN_100 in dependence on Fe concentration, number of trials *n* = 5, means with 95% CI.

**Figure 7 ijms-27-00150-f007:**
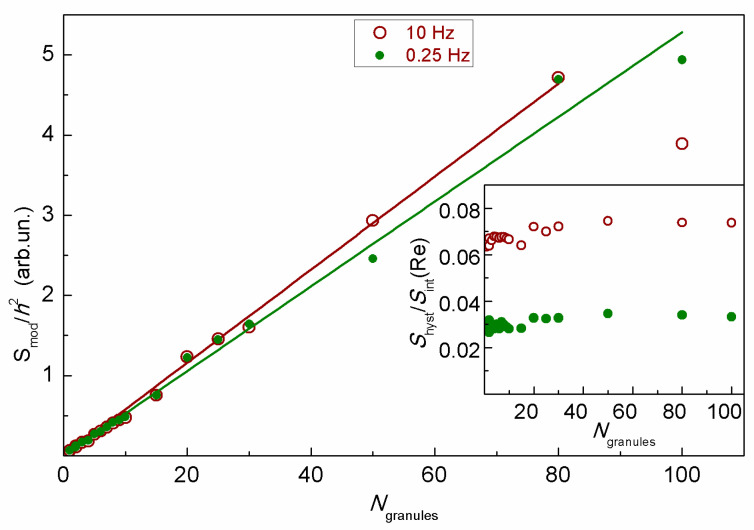
The moduli of the integral recorded signals from the suspension of MSCNP_700 depending on the number of microspheres at two scan frequencies. Inset presents the relative square of the hysteresis loop as a function of the number of microspheres in the suspension for the two scan frequencies. Straight lines in the figure are linear approximations.

**Figure 8 ijms-27-00150-f008:**
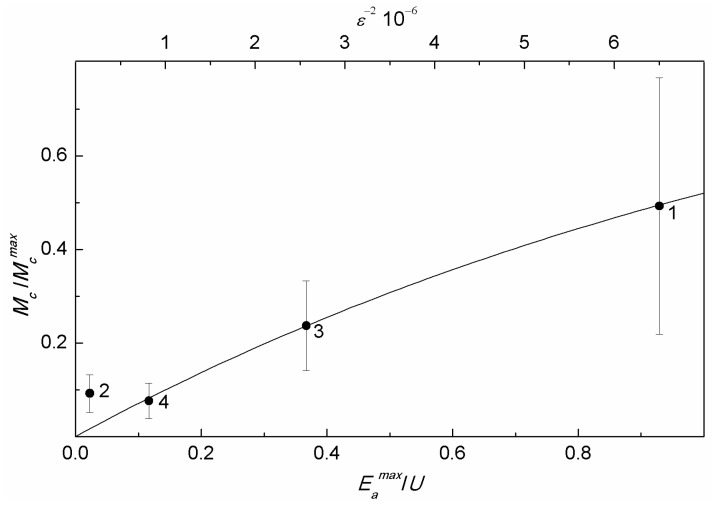
Normalized mean magnetic moment as a function of the ratio $R=E_{a}^{max}/U$ (bottom scale) and *ε^−2^* (top scale) (see text). The curve is best fit. The numbers at the points correspond to Table 1 and Figure 1b.

**Figure 9 ijms-27-00150-f009:**
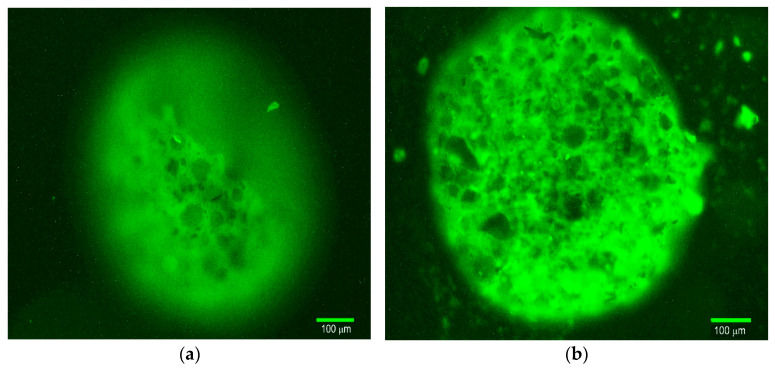
Fluorescence microscopy of TKD-FITC@MSCMN_700: (**a**) conjugated microsphere and (**b**) conjugated microsphere after mechanical destruction. Magnification 20×, scale bar 100 μm.

**Figure 10 ijms-27-00150-f010:**
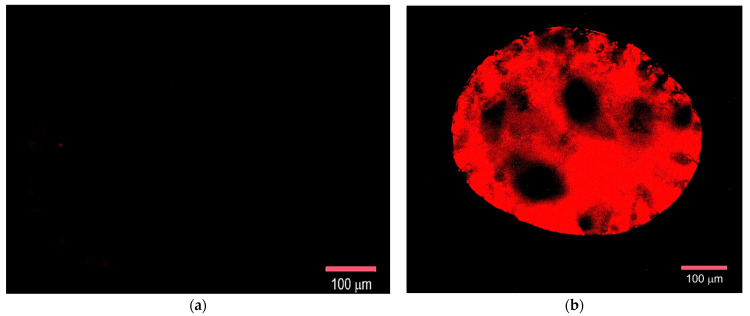
Fluorescence microscopy of MSCMN samples in the red channel after incubation with Hsp70-Flamma648: (**a**) All@MSCMN and (**b**) TKD@MSCMN. Magnification 20×, scale bar 100 μm.

**Figure 11 ijms-27-00150-f011:**
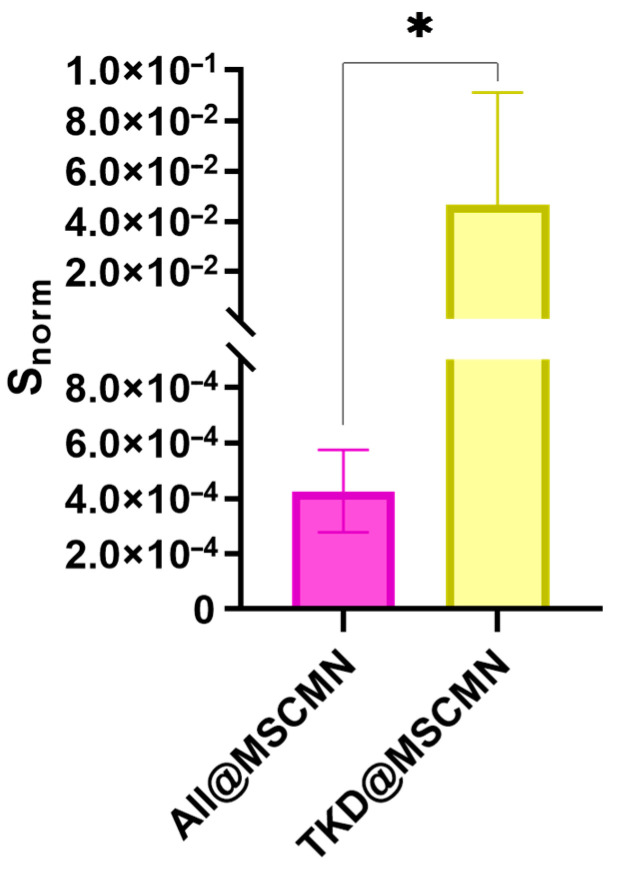
Dependence of the S_norm_ value on the sample type during co-incubation of All@MSCMN and TKD@MSCMN with the Hsp70-Flamma648, number of trials *n* = 10, means with 95% CI. “*” indicates statistical significance at *p* < 0.05.

**Figure 12 ijms-27-00150-f012:**
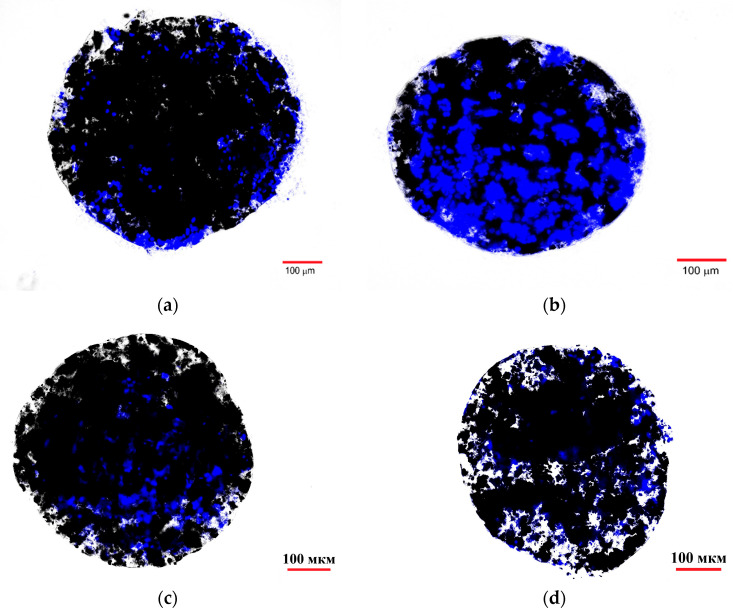
Fluorescence microscopy of MSCMN samples in the blue channel after incubation: (**a**) All@MSCMN with C6 cells; (**b**) TKD@MSCMN with C6 cells; (**c**) TKD@MSCMN with FetMSC cells; and (**d**) MSCMN with C6 cells.

**Figure 13 ijms-27-00150-f013:**
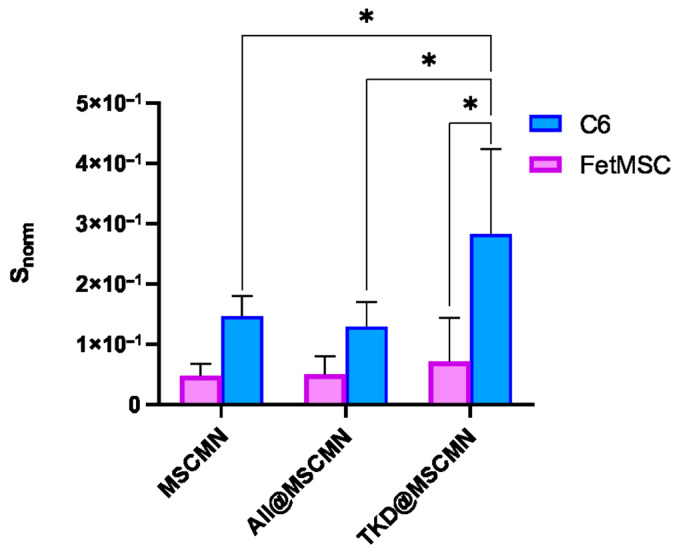
Dependence of the S_norm_ value on the sample type during co-incubation of MSCMN, All@MSCMN and TKD@MSCMN with the C6 cell line, and FetMSC normal cells; number of trials *n* = 10, means with 95% CI. “*” indicates statistical significance at *p* < 0.05.

**Figure 14 ijms-27-00150-f014:**
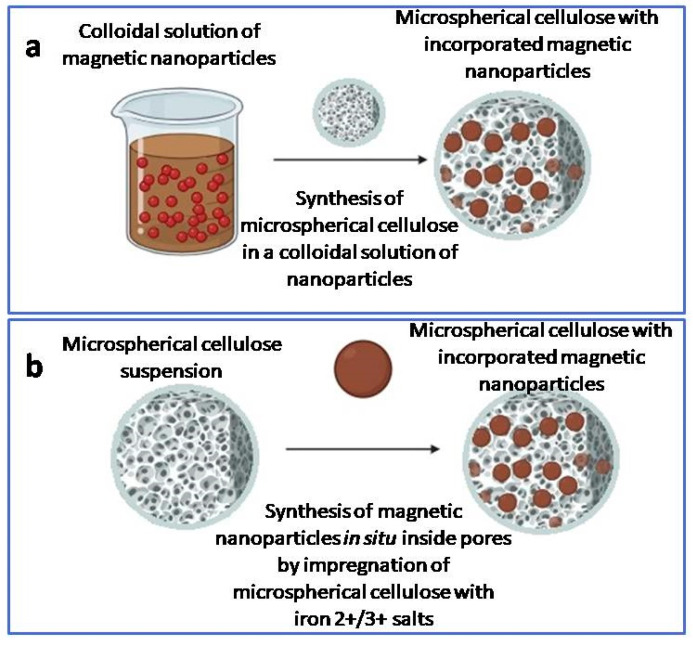
Scheme of synthesis of magnetic microspherical cellulose using pre-prepared iron oxide nanoparticles (**a**) and with the synthesis of iron oxide nanoparticles inside pre-prepared cellulose microspheres (**b**).

**Table 1 ijms-27-00150-t001:** Parameters obtained from X-ray diffraction. The numbers of the samples in the first column correspond to Figure 1b.

Type of Sample		*κλ/d*	*d, nm*	*ε*
1	MSCMN_100	0.0128	10.8(2.0)	0.00039
2	MSCMN_700	0.0081	17.1(2.5)	0.0025
3	MNP_no-shell	0.0098	14.1(1.9)	0.00062
4	MNP_Dx	0.0096	14.4(2.4)	0.0011

**Table 2 ijms-27-00150-t002:** Parameters of magnetic centers in MSCMN_100 at different iron concentrations measured at the scan frequency *F*_sc_ = 0.25.

*C_Fe_*, μg/mL	*N*_C_, 1/mL	*M*_C_, μ_B_	*σ*	α	*τ*_N_, ns
109	2.31(41) × 10^13^	21,460	0.787(16)	0.289(11)	0.483(32)
214	8.59(51) × 10^13^	18,890	0.828(18)	0.269(16)	0.459(22)
328	1.18(13) × 10^14^	18,870	0.841(18)	0.273(14)	0.449(11)
503	2.55(23) × 10^14^	17,320	0.868(20)	0.270(14)	0.418(18)
538	3.14(28) × 10^14^	19,260	0.834(35)	0.267(10)	0.468(30)

**Table 3 ijms-27-00150-t003:** Parameters of magnetic centers in some MSCMN_700 samples with different numbers of microspheres obtained at *F_sc_* = 0.25 Hz.

*N* _MSCMN_	*N* _C_	*M*_C_, μ_B_	*σ* _M_	α	*τ*_N_, ns
1	1.39 × 10^12^	13,850	0.861	0.274	0.333
10	1.16 × 10^13^	13,230	0.868	0.267	0.321
20	2.83 × 10^13^	13,750	0.847	0.276	0.323

## Data Availability

The original contributions presented in this study are included in the article material. Further inquiries can be directed to the corresponding authors.

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
