# Peer review of "Cellulose-Encapsulated Magnetite Nanoparticles for Spiking of Tumor Cells Positive for the Membrane-Bound Hsp70"

_ijms, 2025, doi:10.3390/ijms27010150_

Round 1

Reviewer 1 Report

Comments and Suggestions for Authors

The present manuscript describes the development and characterization of magnetic cellulose microspheres (Cel-m_spheres) functionalized with a TKD peptide for the specific isolation of tumor cells expressing membrane-bound Hsp70 (mHsp70). The main contributions are the synthesis and physico-chemical characterization (XRD, TEM, confocal microscopy) of two types of magnetic microspheres, followed by a demonstration of their specific binding to both the Hsp70 protein and mHsp70-positive C6 glioma cells. A key advantage of the work is the comprehensive approach used to characterize the magnetic nanoparticles, both in suspension and encapsulated within the cellulose matrix, providing a solid base for their proposed biomedical application.

However, after examining the manuscript I have following questions and suggestions:

1) The primary evidence for successful tumor cell isolation is based on semi-quantitative fluorescence microscopy (normalized stained area). This does not provide a direct measure of capture efficiency (the percentage of cells recovered from a known input number), which is crucial for evaluating the system performance for diagnostic applications. Please, perform experiment where a known number of C6 cells are spiked into a buffer or a complex fluid (diluted blood plasma). After incubation with TKD@MSCMN and magnetic separation, quantify the number of bound cells (it can be made by different ways, for example https://doi.org/10.1021/acsomega.4c04154).

2) Specificity is currently demonstrated only by comparing the target TKD-conjugate to a non-specific peptide (Alloferon) conjugate on the same mHsp70-positive C6 cell line. While this is a good control, it does not rule out non-specific adhesion of cells to the cellulose matrix itself. Please, include experiment using a cell line known to have low or undetectable levels of mHsp70. Demonstrate that TKD@MSCMN shows significantly lower binding to this negative control cell line compared to the C6 cells.

3) The descriptions of the synthesis methods for the microspheres (4.2, 4.3, 4.4) are complex and difficult to follow. Please, add a scheme that shows two main synthesis pathways: loading pre-formed MNPs and in-situ synthesis of MNPs within pre-formed cellulose spheres. Re-structure the text to use clearer steps for key procedures, specifying main parameters.

4) There is a contradiction between the Methods section (4.15), which states the use of the Mann-Whitney test, and the Results text (Pages 13-14), which refers to Student's t-test. Correct this throughout the manuscript. Specify which test was used for each analysis. Ensure all p-values are reported accordingly.

5) The discussion of future use in hemodialysis (Page 15) is intriguing but very brief. In the conclusion or discussion, please concentrate on the practical workflow. A sentence or two on the possible protocol for processing a blood sample would enhance the translational impact of the work.

6) Section 4.15 states "Experimental data are expressed as the mean ± the confidence interval (95% CI)." It is more standard to show Mean ± SD or SEM in bar graphs. Please clarify and ensure the error bars in Figures 10 and 12 are correctly labeled.

Finally, manuscript presents well-executed study with a good materials characterization component and a promising biological application. However, addressing the critical comments (particularly regarding the quantitative assessment of cell capture efficiency and the inclusion of a more rigorous specificity control) is essential to fully support the authors’ claims and ensure the manuscript impact. So, major revision of the manuscript is required.

Reviewer 2 Report

Comments and Suggestions for Authors

Cellulose encapsulated magnetite nanoparticles for spiking of tumor cells positive for the membrane-bound Hsp70

The introduction is very detailed, and the references are adequate; however, the text feels overloaded with detail. Please clarify whether the use of C6 rat glioma cells is simply a model, or whether this separation method using TDK@MSCMN nanoparticles could potentially be used for isolating tumor cells.

This section would be clearer if it were shortened and a visualization of the nanoparticles being developed was provided.

At the end of the section, the objective of the study is stated: to investigate the feasibility of producing magnetic cellulose nanoparticles functionalized with the TKD peptide for separating tumor cells. Two additional objectives related to the specifics of nanoparticle synthesis were also formulated. It would be better if you visualized these nanoparticles as a diagram.

Materials and Methods:

Please provide a summary table with all the nanoparticles you used in your study, and describe their size and composition. When reading an article, it's very difficult to constantly search for nanoparticle characteristics in the text.

Describe what the 14-peptide looks like in a 3D configuration? Are there any models available? What percentage of the nanoparticle is covered by the peptide after ligation?

Section 4.8. - Describe your sample preparation.

Please clarify whether it is technically possible to synthesize these nanoparticles in a sterile environment. Was the presence of endotoxin in the nanoparticle suspension assessed?

Section 4.14. Was the viability of the C6 cell line assessed? What number of authors were used for the experiments?

Results:

Table 1 - label the columns, MSCMN_100, and so on.

Figure 6 - clarify n (number of repeats), how are the values ​​represented - median, mean?

Item 2.3 - why do we only see TEM for MSCMN_700?

Figure 10 - clarify n (number of repeats), how are the values ​​represented - median, mean?

Figure 11 - unfortunately, the figure is not representative. Perhaps it would make sense to digitize the pixels and present the data as numerical values.

Figure 12 - clarify n (number of repeats), how are the values ​​represented - median, mean?

 So, we see an interesting and technologically advanced study that has great potential for implementation. However, the article could be improved and made more readable with a more logical and concise introduction. It would be better if you provided a graphical abstract, and it would also be better if you somehow visualized your nanoparticles.

Reviewer 3 Report

Comments and Suggestions for Authors

This manuscript reports synthesis and characterization of cellulose based magnetic particles that could be functionalized with a peptide for binding to tumor cells. The strengths of this work mainly lie in the synthesis and subsequent characterization of varied particles, and may be of interest to the relevant community. I can recommend its publication with revisions.

Introduction is too long and does not get to the main goals in a precise manner. It is much longer than Discussion which is the key part of a research paper. One has to sift through details which should have been placed in Discussion and are unnecessarily distracting.

Results section is mundane, keeps referring to Section 4 and does not contribute towards understanding until one gets to Discussion. Results should be combined with Discussion, and this section should be elaborated further for more comprehensive understanding of the reported work.

Why does the crystallinity decrease during synthesis of the IONPs inside microspheres using CsCl? This certainly plays an important role in their properties including magnetic moment, nanoparticle magnetization etc., and one wonders if it is possible to control/optimize crystallinity (and thus the intended properties) of IONPs?

Minor: There are several grammatical errors which authors could carefully look through. For example:

“Comparison MSCMN_100 and MSCMN_700 parameters from XRD (Fig. 1)….”

Reviewer 4 Report

Comments and Suggestions for Authors

The study demonstrates the clinical significance of creating sensitive tumor cell detection techniques. The study's explicit connection to cancer diagnosis and treatment monitoring effectively frames its biomedical significance. The authors used several techniques, such as confocal microscopy, XRD, TEM, and NLR-M2, demonstrating a multimodal characterization of the microspheres. • The use of the NLR-M2 nonlinear magnetic response technique is clearly novel in this work; additionally, the targeting of mHsp70 with TKD peptide highlights a particular and clinically relevant targeting strategy.

However, there is constructive criticism provided here that will help the authors improve their work.

  1. The abstract is excessively detailed and lengthy. Many of the sentences are complex and contain technical details that are superfluous for an abstract.
  2. The abstract contains no quantitative findings. After spending a lot of time on physical characterization, the abstract quickly shifts to biological functionality. Additionally, readability would be improved by better balance or more obvious transitions.
  3. The introduction, which covers many topics in great detail, is excessively lengthy (many pages). Unrelated side facts are included in some paragraphs, which could detract from the main point. The length of some paragraphs (20–30 lines) makes the text visually and mentally taxing.
  4. Citations support many claims, but in some places, lengthy citation lists disrupt the flow. On the other hand, specific references would be beneficial for some statements (such as the behavior of water in porous cellulose).
  5. Several paragraphs (particularly 2.1, 2.5, and 2.6) describe experimental procedures that should be in Methods rather than Results.
  6. For binding quantification, fluorescence microscopy, and TEM: There is no n (number of microspheres, pictures, or cells examined) given. There is no information regarding the number of measurements represented by the histograms.
  7. No variance or error bars, as in Figures 10 and 12.
  8. Inadequate Imaging Analysis Description
  9. No more than one example of representative negative control images
    There may be a need for additional images or replicates.
  10. Different binding efficiencies cannot be fully explained mechanistically.
    You clarify that the Hsp70-dye complex can penetrate pores.
    However, the distribution of internal and surface peptides is not quantitatively compared. There is no pore-level peptide localization visualization (no 3D reconstructions or confocal z-stacks). There is no proof that the internal peptide works.
    11. There is no clear relationship between the XRD/TEM/NLR-M2 results
  11. From the Stability of the peptide attachment point of view, you do not report the Stability under physiological conditions, whether the peptide desorbs or hydrolyzes over time.
  12. Without visible scale bars, axis labels, or legends, the descriptions in Figures 3–7 are inadequate.
  13. Microsphere pore structure and size distribution: No measurement technique is provided, despite the statement that "internal pore volume ≥ 90%".
  14. The ratio of microspheres to cells is unknown. The number of microspheres added, the weight/volume ratio (µg microspheres per 10² cells), and whether the microspheres were sterilized are all missing from the incubation of cells with MSCMN samples.
  15. Inconsistent Terminology and Abbreviations: MSCMN is sometimes written as MSCNP, MSCMH, or MSCMN-NHâ‚‚. The definition of abbreviations is not always the same.
  16. The primary findings are not quantitatively stated in the conclusion.
    Instead of summarizing the main numerical results or their significance, the section primarily restates the Methods (e.g., nanoparticle synthesis with/without CsCl, XRD analysis, NLR-M2 method).
  17. The biological significance is not discussed in the conclusion. You claim that TKD@MSCMN "recognizes and binds to C6 cells," but what was the quantitative strength of the binding in comparison to the control? Does the technique function in populations with mixed cells?
Comments on the Quality of English Language

Required Polishing

Round 2

Reviewer 1 Report

Comments and Suggestions for Authors

The authors have addressed all critical points raised by me in the initial review comprehensively and satisfactorily.

The manuscript now presents a well-executed, novel, and scientifically sound study that makes a good contribution to the fields of nanobiotechnology, affinity cell separation, and liquid biopsy development. It is suitable for publication in the International Journal of Molecular Sciences following final minor editorial corrections to resolve the formatting and language issues present in the current draft.

Reviewer 2 Report

Comments and Suggestions for Authors

The authors have submitted a revised version of the article, addressing all comments and answering all questions. Specifically, the introduction is now more logical, the objective of the study is stated at the end of the section, all figures have captions, a table with nanoparticle characteristics has been created, and a summary scheme for nanoparticle synthesis has been provided. Overall, the new version of the article is more understandable to the reader.

Reviewer 3 Report

Comments and Suggestions for Authors

Authors have largely addressed the concerns raised earlier. I would suggest grammatical run through during final editorial processing of the accepted manuscript. 

Reviewer 4 Report

Comments and Suggestions for Authors

The authors addressed most of the comments raised during the revision process. Therefore, the paper can be accepted.